# Structure Identification of Two Polysaccharides from *Morchella sextelata* with Antioxidant Activity

**DOI:** 10.3390/foods11070982

**Published:** 2022-03-28

**Authors:** Feng Li, Yu Jin, Jun Wang, Huaide Xu

**Affiliations:** College of Food Science and Engineering, Northwest A&F University, Yangling, Xianyang 712100, China; lifeng90@nwafu.edu.cn (F.L.); jinyu3518@126.com (Y.J.); jun.wang@nwafu.edu.cn (J.W.)

**Keywords:** *Morchella sextelata* polysaccharides, structure, conformation, antioxidant activity

## Abstract

Mushrooms of the *Morchella* genus exhibit a variety of biological activities. Two polysaccharides (MSP1-1, 389.0 kDa; MSP1-2, 23.4 kDa) were isolated from *Morchella sextelata* by subcritical water extraction and column chromatography fractionation. Methylation and nuclear magnetic resonance analysis determined MSP1-1 as a glucan with a backbone of (1→4)-α-D-glucan branched at O-6, and MSP1-2 as a galactomannan with coextracted α-glucan. Light scattering analysis and transmission electron microscopy revealed that MSP1-1 possessed a random coil chain and that MSP1-2 had a network chain. This is the first time that a network structure has been observed in a polysaccharide from *M. sextelata*. Despite the differences in their chemical structures and conformations, both MSP1-1 and MSP1-2 possessed good thermal stability and showed antioxidant activity. This study provides fundamental data on the structure–activity relationships of *M. sextelata* polysaccharides.

## 1. Introduction

Natural polysaccharides have attracted extensive attention because of their advantages, such as safety, biocompatibility, and in vivo biodegradability. Edible mushrooms are rich sources of natural polysaccharides. Mushroom polysaccharides have wide applications in the food industry as dietary supplements, nutritional promoters to flour and dairy products, or edible films for packaging materials [1].

Polysaccharides possess high capacities for carrying biological information because of their large structural variability. Chemically, most mushroom polysaccharides are glucans (α-, β-, or α/β) or heteropolysaccharides. However, polysaccharides with the same chemical structure can exhibit different biological activities after changes in conformation [2]. Information on the chemical composition, molecular weight (MW), primary structure (monosaccharide composition, skeleton structure, and branching structure), and advanced structure (three-dimensional chain conformation) of polysaccharides is important to understanding their activities. Therefore, the first step toward elucidating structure–activity relationships is to determine the molecular structure.

*Morchella* spp., commonly known as morel, are edible and medicinal mushrooms that are valued for their unique flavor, and are categorized as morphologically recognizable mushrooms. In particular, *Morchella sextelata* belongs to the family of Morchellaceae, and has been successfully cultivated and commercialized in China. Morel contains polysaccharides, proteins, phenolics, and sterol derivatives [3], with polysaccharides being considered as the most effective substance that endows morel with anti-inflammatory effects [4] to enhance immunity [5] and reduce the risk of tumors and cancer [6,7]. Polysaccharides isolated from morel are reported to have backbones of 4)-α-D-Glc*p*-(1→, →4)-α-D-Glc*p*-(1 →6)-α-D-Gal*p*-(1→, →2)-α-D-Gal*p*-(1→3,6)-α-D-Man*p*-(1→2,3,6)-α-D-Man*p*-(1→ [5,8,9,10]. However, the fine structure of the polysaccharides in *M. sextelata* has rarely been investigated. Our previous study demonstrated that polysaccharides could be extracted from *M. sextelata* using subcritical water, and that the extract (MSP_SWE_) had potent antioxidant and immune activities [11]. This study aimed to obtain the structural features of purified MSP_SWE_. Moreover, the in vitro antioxidant activities of polysaccharides were evaluated. These results will be helpful for elucidating the structure–activity relationships of *M. sextelata* polysaccharides and promoting their application in functional foods.

## 2. Materials and Methods

### 2.1. Materials and Pretreatment Methods

Cultured *M. sextelata* was collected from Hubei (China). The dried fruiting body of *m. sextelata* was pulverized, filtered through an 80-mesh sieve, and preserved in a drying dish.

### 2.2. Fractionation of M. sextelata Polysaccharide

*M. sextelata* polysaccharide (MSP_SWE_) was extracted via subcritical water extraction as described previously [11]. The MSP_SWE_ (20 mg/mL) was introduced into a DEAE Sepharose FF (2.6 × 60 cm) column and subsequently washed with running distilled water (flow rate: 1.2 mL/min; liquid volume: 6 mL/tube). The carbohydrate content in the tubes was monitored. Then, the main fraction (MSP1) was combined and freeze-dried. MSP1 (30 mg) was fractionated further by gel filtration chromatography on a Sepharose CL-6B (1.6 × 90 cm) column. The sample (6 mg/mL) was applied to the column, eluted with distilled water (flow rate: 0.3 mL/min; 3 mL/fraction), and the two fractions obtained were labelled as MSP1-1 and MSP1-2.

### 2.3. General Analysis of MSP1-1 and MSP1-2

The average MWs of the samples were determined by high-performance gel filtration chromatography (HPGFC). Each sample (50 mg) was dissolved in 10 mL 0.1 M NaNO_3_, and the sample solution was analyzed using Waters 2695 high-performance liquid chromatography (HPLC) system with an Ultrahydrogel™ linear (300 mm × 7.8 mm id) gel-filtration column at 40 °C. The MWs of MSP1-1 and MSP1-2 were estimated based on the standard curve (Lg MW = −0.517*x* + 13.3, *R*^2^ = 0.9953; *x*: retention time) acquired using standard dextrans (T-300, T-150, T-10, and T-5 and glucose).

Ultraviolet-visible (UV–vis) spectroscopy was performed on the purified polysaccharides (1.0 mg/mL each) using a Shimadzu UV-2550 spectrophotometer (Shimadzu, Japan). The scan-related parameters were set as scan interval 0.5 nm and scan range 200–400 nm.

The functional groups of MSP1-1 and MSP1-2 were analyzed using a Bruker Vertex 70 (Billerica, MA, USA) Fourier transform infrared (FTIR) spectrometer with a vibration region of 4000–400 cm^−1^. Polysaccharides were incorporated with KBr (1:100) for analysis.

Constituent sugar analysis of the polysaccharide fractions was conducted using an ion chromatography system (ICS-5000) equipped with a DionexCarbopac^TM^ PA20 (150 mm × 3.0 mm id) column and pulsed amperometric detector (PAD). Each sample (10 mg) was precisely weighed and hydrolyzed for 3 h with trifluoroacetic acid (3 M, 10 mL) at 120 °C. The acid hydrolysate was dried under nitrogen and mixed thoroughly with 5 mL distilled water. The mixture (100 µL) was then drawn into deionized water (900 µL) and centrifuged. The supernatant (5 µL) was analyzed by ICS-5000 and eluted with a mobile phase (H_2_O, 15 mM NaOH, and 100 mM sodium acetate solution) at 30 °C and a set elution flow rate of 0.3 mL/min.

### 2.4. Methylation

The samples were methylated using the method described previously [12]. MSP1-1 and MSP1-2 were permethylated according to the following steps: first, 10 mg of MSP1-1 or MSP1-2 was weighed and dissolved in 500 μL of DMSO, and 50 μL of DMSO/NaOH solution (120 mg/mL) was added to the sample solution to initiate a reaction. After 30 min, iodomethane (50 μL) was added and reacted for 1 h. The reaction was terminated by the addition of deionized water (1 mL). The product was subsequently extracted using dichloromethane (500 μL) and dried with nitrogen. Trifluoroacetic acid (2 M) was added to the product after methylation was completed, and a hydrolysis procedure was performed (reaction at 121 °C for 90 min). Finally, the obtained product was reduced with 2 M ammonia and 1 M NaBD_4,_ and acetylated with acetic anhydride. To detect the ultimate derivatives, an Agilent 7890A-5977B gas chromatography–mass spectrometry (GC-MS) instrument (Santa Clara, CA, USA) with a BPX70 (25 m × 0.22 mm × 0.25 μm, Trajan) was employed with high-purity helium as carrier gas at an injection volume of 1 μL. The temperature of the BPX70 column was initially 140 °C (held for 2.0 min) and then ramped up to 230 °C (held for 3 min) using a 3 °C/min program. The peaks of the derivatives were identified from the relative retention times and mass spectral data.

### 2.5. Nuclear Magnetic Resonance (NMR) Spectroscopy

The NMR data were obtained using a 500 MHz Bruker NMR spectrometer (Zurich, Switzerland) at 25 °C. MSP1-1 and MSP1-2 (20 mg each) were dissolved in D_2_O solvent and lyophilized. The dried polysaccharides (20 mg) were transferred to a standard 5 mm probe after dissolution in 500 μL D_2_O solvent. The operating frequencies of ^1^H NMR and ^13^C NMR were 500 and 125 MHz, respectively. The acquisition times were set to 16 times for ^1^H NMR spectra, 1024 times for ^13^C NMR spectra, 12 times for ^1^H/^1^H correlation spectroscopy (COSY), 12 times for heteronuclear single quantum correlated spectroscopy (HSQC), 32 times for nuclear overhauser effect spectroscopy (NOESY), and 64 times for heteronuclear multiple bond correlation (HMBC) spectra. MestReNova software (Version: 9.1.0-14011; Mestrelab Reserch, Santiago de Compostela, Spain) was used to process the NMR data of the polysaccharide samples.

### 2.6. Congo Red Test

The presence of helical structures in the polysaccharide samples was tested using a method described by Jia et al. [13]. That is, the sample solution (1.0 mg/mL, 1.0 mL), Congo red (80 mΜ, 1.0 mL), and NaOH (1.0 M) were mixed until the NaOH concentration in the mixed solution gradually increased to 0.5 M. The blank control was a Congo red solution without polysaccharides. The UV-2550 spectrometer was run to scan the maximum absorption wavelength (*λ*_max_) at each NaOH concentration.

### 2.7. Light Scattering Measurements and Morphological Observations

The polysaccharide samples were dissolved in 0.9% NaCl solution (3 mg/mL) for conducting dynamic light scattering (DLS) measurements using a Brookhaven light scattering spectrometer (BI-200SM, Brookhaven, GA, USA) at a wavelength of 532 nm and a 90° angle. The sample solutions were diluted to varying concentrations (0.1–0.75 mg/mL) by using 0.9% NaCl solution and stirred for 24 h after each dilution. The solutions were measured at angles of 50°–140° for static light scattering (SLS). All solutions were purified (filter: 0.22 μm) three times before testing. The obtained data were fitted using Berry plots.

The chain conformation of the polysaccharides was assessed using a Tecnai G2 Spirit Bio transmission electron microscope (TEM; FEI, Portland, OR, USA) at 80 kV. MSP1-1 or MSP1-2 was dissolved in 0.9% NaCl solution to achieve a final concentration of 5 μg/mL. A drop of the polysaccharide solution was deposited on the support carbon film (200 mesh), dried in the air, and stained with 0.2% phosphotungstic acid.

### 2.8. Thermal Analysis

Simultaneous thermogravimetric and differential scanning calorimetry (TG-DSC) was performed using an STA 449 F3 thermogravimetric analyzer (Netzsch, Selb, Germany). The samples (5 mg each) were sealed in an Al_2_O_3_ crucible and heated from 25 °C to 600 °C (heating rate: 10 °C/min) under a nitrogen atmosphere.

### 2.9. Antioxidant Activity

The antioxidant activity (DPPH radical scavenging and ferric reducing activity) test of the polysaccharides was measured as described previously [11]. The specific procedures are supplied in Appendix A.

### 2.10. Statistical Analysis

All data were obtained using three independent trials, and results were expressed as mean ± standard deviation. One-way analysis of variance (ANOVA) was performed using SPSS 18.0 software. The Tukey test results were significantly different in cases where the *p*-values were less than 0.05.

## 3. Results and Discussion

### 3.1. Structural Characterization of MSP1-1 and MSP1-2

MSP_SWE_-derived MSP1-1 (43.3%, *w*/*w*) and MSP1-2 (18.2%, *w*/*w*) were separated as shown in Appendix A. The total carbohydrate contents of MSP1-1 and MSP1-2 were 98.0% ± 0.8% and 98.7% ± 0.7%, respectively. The extracts showed no absorption at 260–280 nm, confirming that MSP1-1 and MSP1-2 were free of nucleic acids and proteins (shown in Appendix A).

The HPGFC profiles of MSP1-1 and MSP1-2 (Figure 1A) both yielded single peaks, indicating their homogeneity. The MWs of MSP1-1 and MSP1-2 were calculated as 389 and 23.4 kDa, respectively, based on the retention times of dextran standards. Constituent sugar analyses showed that MSP1-1 contained only glucose, whereas MSP1-2 contained mostly glucose (46.0 mol%), mannose (37.5 mol%), and galactose (16.0 mol%), with a small amount of glucosamine (0.5 mol%) (Figure 1B).

The FT-IR spectra of MSP1-1 and MSP1-2 exhibited the characteristic peaks of general polysaccharides (Figure 2). The common bands emerged at 3400, 2931, 1640, and 1406 cm^–1^, representing the absorptions of the O–H, C–H, associated water, and C–H groups, respectively [14]. The C–O–C and C–O–H groups in the pyran structure generated typical peaks at 1157, 1080, and 1024 cm^–1^ [15]. MSP1-1 exhibited unique absorptions at 931, 854, and 763 cm^–1^ related to α-glucans [16]. In comparison, an additional absorption peak at 813 cm^–1^ in MSP1-2 indicated the presence of mannose [17].

### 3.2. Methylation Analysis of MSP1-1 and MSP1-2

The GC-MS results for the methylation analysis are shown in Table 1 and Appendix A. Four linkage mechanisms were identified for MSP1-1: t-Glc(*p*), 6-Glc(*p*), 4-Glc(*p*), and 4,6-Glc(*p*); and eight linkage mechanisms for MSP1-2: t-Glc(*p*), t-Gal(f), 2-Man(*p*), 6-Man(*p*), 5-Gal(*f*), 4-Glc(*p*), 2,6-Man(*p*), and 4,6-Glc(*p*). The degree of branching (DB) values of MSP1-1 and MSP1-2 were calculated as 28.3% and 34.6%, respectively, according to the method reported by Hawker and Lee [18].

### 3.3. NMR Analysis of MSP1-1 and MSP1-2

Detailed structural information regarding MSP1-1 and MSP1-2 was elucidated based on the chemical shifts in the NMR spectra. The ^1^H NMR spectrum of MSP1-1 (Figure 3A) reveals a large number of resonance signals of overlapping protons in the δ 3.0–5.5 ppm region. The signals appear in the anomeric region at δ 5.35, δ 5.31, δ 4.92, δ 4.92, δ 5.18, and δ 4.58 ppm, and are denoted by A, B, C, D, R_α_, and R_β_, respectively. In the ^13^C spectrum of MSP1-1 (Figure 3B), the anomeric carbon signals of residues A, B, C, D, R_α_, and R_β_ were determined to be at δ 100.0, δ 100.0, δ 98.5, δ 99.9, δ 92.0, and δ 95.8 ppm, respectively.

After the origin of the anomeric signal was determined, the NMR spectra and methylation results were compared with the reported data for similar glycosyl substitutions [5,19]. All chemical shifts were assigned to the residues of MSP1-1 and given in Table 2.

The HMBC spectrum (Figure 3E) reveals the following correlations: A H-1/A C-4, A H-1/B C-4, and A H-4/A C-1. The linkages of the polysaccharide samples are further verified by the NOESY spectrum (Figure 3F). Results of the methylation and NMR analyses indicate that MSP1-1 is composed of residue A through a 1→4 glycosidic bond, and that residues A and B are connected to form the sugar backbone. On the branch, residues B and C are connected to residue D through position 6. The possible repeating structural units of MSP1-1 are shown in Figure 3G.

Similarly, as shown in Figure 4A–F, the spectra of MSP1-2 reveal eight residues with signals at δ 5.20/100.9, δ 5.02/102.3, δ 5.08/98.4, δ 4.93/98.7, δ 5.36/99.7, δ 5.31/99.9, δ 5.34/108.4, and δ 5.18/109.5 ppm labeled as A–H. The monosaccharide composition, methylation analysis results, and chemical shifts of the abovementioned anomeric protons and anomeric carbons indicate that residues A–F possess α-configurations, and that residues G and H are present in β-configurations [20,21]. The COSY, HSQC, HMBC, and 1D spectra and methylation results were compared with the reported data for similar glycosyl substitutions [20,22], and the corresponding ^1^H and ^13^C chemical shifts in MSP1-2 are given in Table 2.

The HMBC (Figure 4E) and NOESY spectra (Figure 4F) identify sequences between sugar residues in MSP1-2. A cross-peak between D H-1 and F H-6 indicates that sugar residue D is connected to sugar residue F by an α-1,6 glycosidic bond. Observed cross-peak of G H-1/C H-6 implies the existence of G→C linkage. In addition, the interaction between B H-1/C H-6 confirms the B→C linkage. Similarly, the observed cross-peaks of A H-1/B C-6, F H-1/E C-4, and H H-1/G C-5 confirm the A→B, F→E, H→G linkage. Overall, we infer that MSP1-2 is a polymer with a skeleton consisting of 1,2-, 1,6-, 1,2,6-linked α-D-Man*p* residues and (1→4)-α-D-glucose. The possible sequence fragments present in MSP1-2 are shown in Figure 4G.

MSP1-1 and MSP1-2 share similar structures with some polysaccharides displaying unique biological activities. The glucan with a backbone of →4)-α-D-Glc*p*→ has been found in *Cordyceps sinensis* [23], *Agaricus blazei* [24], ginger [25], *Dictyophora echinovolvata* [19], quinoa [26], and *Glycyrrhiza glabra* [27]. The structure of MSP1-1 is closer to that of *Morchella importuna* polysaccharide (MIPB50-S-1) [5], which exhibits immunomodulatory activity. They have the same main chain and branch structure but different DBs and MWs. A polysaccharide consisting of α-mannose and α-glucose in the structure has been reported in *Boletus edulis* [28] and *Hippophae rhamnoides* L. [29]. A polysaccharide isolated from *Boletus edulis* is predominated by α-1, 3-linked Glc*p* and α-1, 4-linked Man*p* residues, which can adsorb heavy metals. A heteropolysaccharide purified from *Hippophae rhamnoides* L. with a backbone of α-1, 4-linked Glc*p*, α-1,4,6-linked Glc*p*, and α-1, 4-linked Man*p* residues have shown an intestinal protective effect. Therefore, further study on MSP1-1 and MSP1-2 may reveal their potential bioactivities.

### 3.4. Conformation and Morphological Properties in Solution

#### 3.4.1. Triple Helix Structural Determination

Congo red solution is an indicator for identifying the triple helix conformations in polysaccharides [30]. With an increasing concentration of NaOH, no characteristic red shift was observed at *λ*_max_ related to the polysaccharide–Congo red complexes (shown in Appendix A), indicating that neither MSP1-1 nor MSP1-2 had a triple helix structure.

#### 3.4.2. Conformational Characteristics

Polysaccharides tend to aggregate in water, and the chain conformation in 0.9% NaCl solution was studied by light scattering (shown in Appendix A). MSP1-1 showed a single particle size distribution with a hydrodynamic radius (*R*_h_) of 28.89 nm. MSP1-2 aggregated in 0.9% NaCl solution and yielded a bimodal size distribution with a *R*_h_ of 34.87 nm. The calculated values of the conformational parameters are presented in Table 3. The second virial coefficient (A_2_) of MSP1-1 is positive, indicating that 0.9% NaCl solution is a good solvent, whereas A_2_ is negative for MSP1-2, showing that aggregates existed in MSP1-2/0.9% NaCl solution [31]. Both MSP1-1 and MSP1-2 had a random coil conformation as their structure parameter ρ = *R*_g_/*R*_h_ values (*R*_g_: radius of gyration) were calculated as 1.89 and 1.46, respectively [32]. 

The chain conformations of MSP1-1 and MSP1-2 were verified using TEM in 0.9% NaCl solution. In Figure 5A, MSP1-1 resembles a random coil chain, which is consistent with the light scattering results. MSP1-2 exhibits a unique fishnet-like conformation (Figure 5B). Similar aggregates have been reported in lentinan polysaccharides, and the driving force for this aggregation was hydrogen bonding [33]. This is the first time that a network structure has been observed in an *M. sextelata* polysaccharide. In this conformation, the polysaccharide may exhibit utility as a carrier.

### 3.5. Thermal Properties

TGA curves indicate two-stage decomposition for both MSP1-1 and MSP1-2 (Appendix A). The first stage of degradation occurred before 100 °C due to the rapid evaporation of free water [34] with a weight loss of 10.49% for MSP1-1 and 8.89% for MSP1-2. The second degradation occurred in the temperature interval of 140–400 °C, and the weight loss of MSP1-1 and MSP1-2 were 74.56% and 82.49%, respectively. This is related to the thermal decomposition of the polysaccharides [35]. The derivative thermogravimetric curves revealed that the maximal degradation of MSP1-1 and MSP1-2 occurred at 308.50 °C and 309.01 °C, respectively. This thermal behavior was similar to that observed for polysaccharides derived from *Ribes nigrum* L. [36] and *Ocimum album* L. seed [37].

The DSC thermograms of MSP1-1 and MSP1-2 were consistent with the TGA results, revealing two marked endothermic transitions attributed predominantly to the dehydration and thermal decomposition of the polysaccharides. Degradation began at 261.70 °C for MSP1-1 with a maximal peak at 305.06 °C, and at 279.69 °C for MSP1-2 with a maximal peak at 315.57 °C. The higher decomposition temperature of MSP1-2 indicates that it has better thermal stability compared to MSP1-1 [38]. Additionally, the higher endothermic enthalpy change (Δ*H* = 263.86 J/g) confirms that MSP1-2 possesses better thermal stability [39], which may be related to its shorter chain.

### 3.6. Antioxidant Activity In Vitro

Compared to the positive control group (Vc), MSP1-1 and MSP1-2 can scavenge free DPPH radicals (Figure 6A). The scavenging activity of samples was correlated with increasing concentration (0.5–4 mg/mL). At a concentration of 4 mg/mL, the scavenging activities of MSP1-1 and MSP1-2 were 43.41% and 58.09%, respectively, suggesting that MSP1-2 has higher antioxidant activity than MSP1-1 (*p* < 0.05). This activity is related to the chemical structure of MSP1-2 because polysaccharides containing glucose and mannose were reported to have good antioxidant activity. Moreover, the higher molecular weight of MSP1-1 affects its scavenging ability [12,40]. The scavenging activity of pure *M. sextelata* polysaccharide was significantly decreased compared with that of crude polysaccharides at the corresponding concentration (*p* < 0.05) [11] because some active substances with antioxidant properties have been removed in the purification process. The ferric reducing abilities of MSP1-1 and MSP1-2 were significantly decreased compared to that of the Vc group (*p* < 0.05). However, they still showed a certain reducing ability, which was higher than that of *Laminaria japonica* polysaccharides at 2.0 mg/mL [41]. In summary, these results showed that *M. sextelata* polysaccharides have antioxidant capacity and may be potentially used as a natural antioxidant.

## 4. Conclusions

In the present study, we successfully isolated two novel polysaccharides (MSP1-1 and MSP1-2) via the subcritical water extraction from *M. sextelata* fruiting bodies. The structure, chain conformation, and thermal stability of these two polysaccharides were investigated using different analytical methods. MSP1-1 displayed random coil chains with a backbone of α-1, 4-linked Glc*p*, which was substituted at the O-6 by a short branch. MSP1-2 showed a network chain with two domains, a backbone of α-1, 2-linked Man*p* and α-1, 6-linked Man*p*, and another α-1, 4-linked Glc*p* backbone. Both polysaccharides showed antioxidant activity with some variations due to differences in their chemical structures and molecular weights. This research outcome will be beneficial to the development of green applications in the food and medical sciences. The antioxidant activity in vivo of *M. sextelata* polysaccharides requires additional investigations.

## Figures and Tables

**Figure 1 foods-11-00982-f001:**
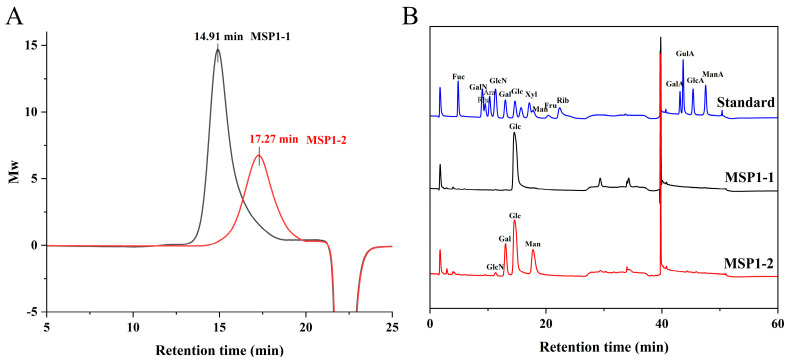
(**A**) High-performance gel filtration chromatography (HPGFC) elution profiles of MSP1-1 and MSP1-2. (**B**) Chromatographic analysis results of mixed monosaccharide standard, MSP1-1, and MSP1-2.

**Figure 2 foods-11-00982-f002:**
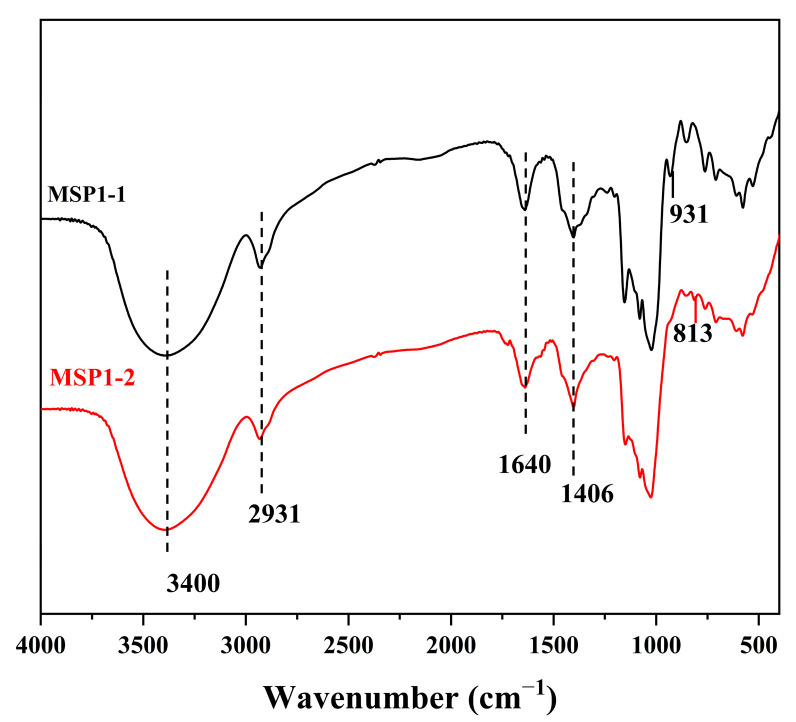
Fourier transform infrared (FT-IR) spectra of MSP1-1 and MSP1-2.

**Figure 3 foods-11-00982-f003:**
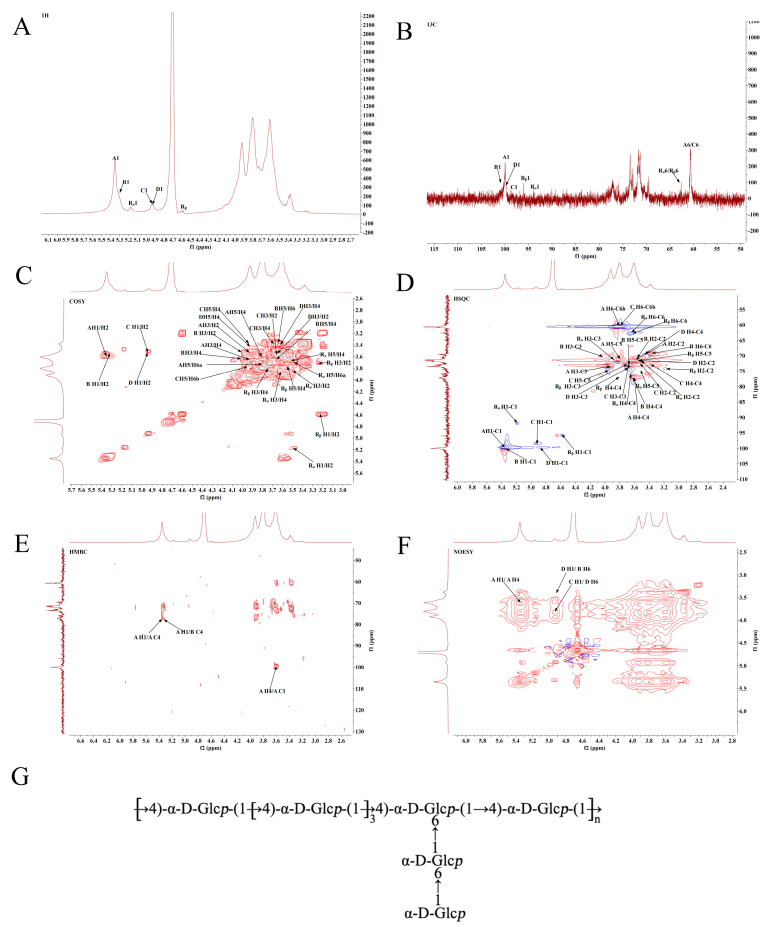
NMR spectra and proposed structures of MSP1-1. (**A**) ^1^H NMR, (**B**) ^13^C NMR, (**C**) ^1^H/^1^H correlation spectroscopy (COSY), (**D**) heteronuclear single quantum correlated spectroscopy (HSQC), (**E**) heteronuclear multiple bond correlation (HMBC), and (**F**) nuclear overhauser effect spectroscopy (NOESY); (**G**) proposed structures of MSP1-1.

**Figure 4 foods-11-00982-f004:**
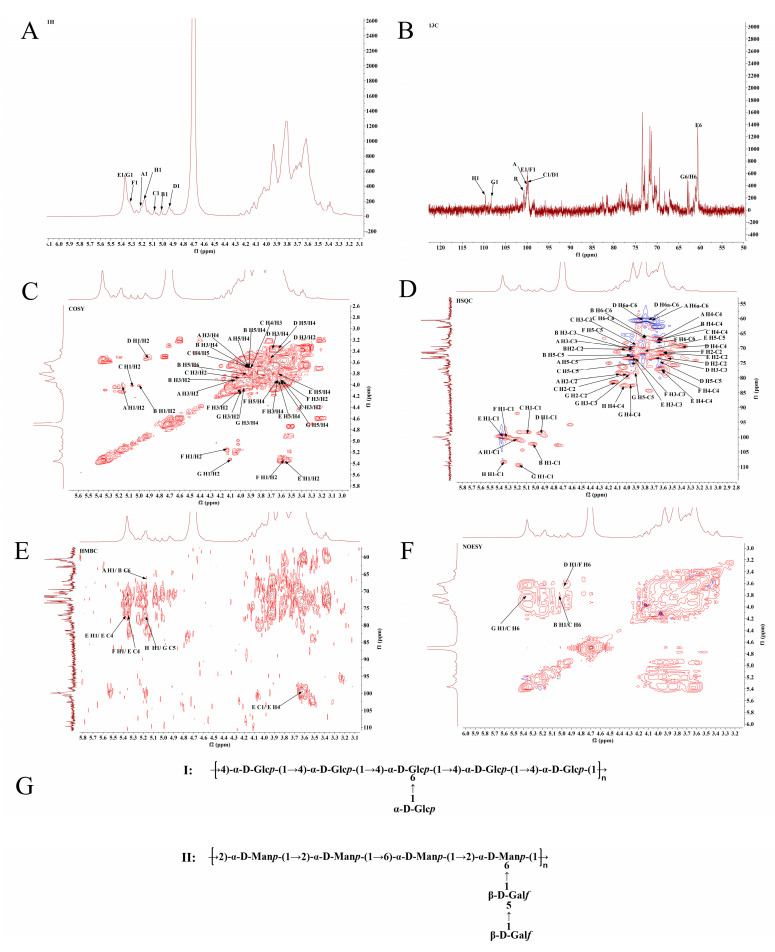
NMR spectra and proposed structures of MSP1-2. (**A**) ^1^H NMR, (**B**) ^13^C NMR, (**C**) COSY, (**D**) HSQC, (**E**) HMBC, and (**F**) NOESY; (**G**) representation of possible structures of MSP1-2, type Ⅰ: galactomannan; type Ⅱ: glucan.

**Figure 5 foods-11-00982-f005:**
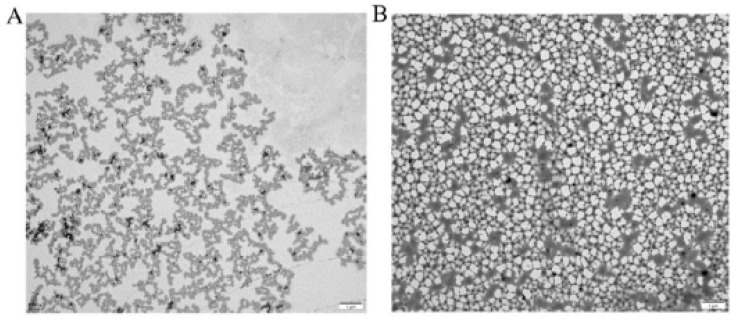
Transmission electron microscopy (TEM) images of (**A**) MSP1-1 and (**B**) MSP1-2 in 0.9% NaCl solution.

**Figure 6 foods-11-00982-f006:**
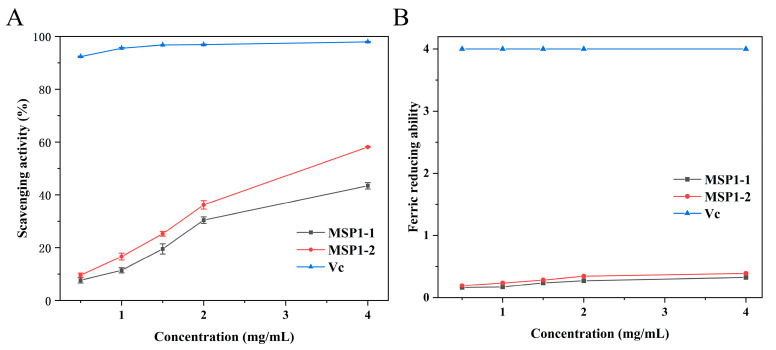
Antioxidant activity of MSP1-1 and MSP1-2. Scavenging activity on DPPH free radical (**A**), and ferric reducing ability (**B**).

**Table 1 foods-11-00982-t001:** Glycosidic linkage type composition of MSP1-1 and MSP1-2 based on gas chromatography–mass spectrometry (GC-MS) results.

Sample	Retention Time (min)	PMAAs	Linkage Pattern	Relative Percentage (%)
MSP1-1	9.3	1,5-di-O-acetyl-2,3,4,6-tetra-O-methyl glucitol	t-Glc(*p*)	15.9
	14.2	1,5,6-tri-O-acetyl-2,3,4-tri-O-methyl glucitol	6-Glc(*p*)	10.9
	14.5	1,4,5-tri-O-acetyl-2,3,6-tri-O-methyl glucitol	4-Glc(*p*)	60.8
	18.8	1,4,5,6-tetra-O-acetyl-2,3-di-O-methyl glucitol	4,6-Glc(*p*)	12.4
MSP1-2	9.4	1,5-di-O-acetyl-2,3,4,6-tetra-O-methyl glucitol	t-Glc(*p*)	10.6
	9.9	1,4-di-O-acetyl-2,3,5,6-tetra-O-methyl galactitol	t-Gal(*f*)	8.0
	12.9	1,2,5-tri-O-acetyl-3,4,6-tri-O-methyl mannitol	2-Man(*p*)	15.1
	14.2	1,5,6-tri-O-acetyl-2,3,4-tri-O-methyl mannitol	6-Man(*p*)	7.7
	14.2	1,4,5-tri-O-acetyl-2,3,6-tri-O-methyl	5-Gal(*f*)	5.9
	14.6	1,4,5-tri-O-acetyl-2,3,6-tri-O-methyl	4-Glc(*p*)	36.7
	18.8	1,2,5,6-tetra-O-acetyl-3,4-di-O-methyl	2,6-Man(*p*)	6.3
	18.9	1,4,5,6-tetra-O-acetyl-2,3-di-O-methyl	4,6-Glc(*p*)	9.7

**Table 2 foods-11-00982-t002:** ^1^H and ^13^C NMR chemical shifts for MSP1-1 and MSP1-2 in D_2_O.

Sample	Sugar Residue		Chemical Shifts (ppm) ^a^							
				1	2	3	4	5	6a	6b
MSP1-1	A	→4)-α-D-Glc*p*-(1→	H	5.35	3.59	3.92	3.60	3.80	3.73	3.80
			C	100.0	71.7	73.4	77.1	71.3	60.7	
	B	→4,6)-α-D-Glc*p*-(1→	H	5.31	3.56	4.01	3.62	3.65	3.37	
			C	100.0	71.6	70.2	77.1	72.7	69.4	
	C	α-D-Glc*p*-(1→	H	4.92	3.52	3.66	3.36	3.95	3.73	3.80
			C	98.5	72.8	72.9	73.0	74.4	60.6	
	D	→6)-α-D-Glc*p*-(1→	H	4.92	3.54	3.64	3.46	3.93	3.77	
			C	99.9	71.3	73.9	69.5	70.8	68.3	
	R_α_	→4)-α-D-Glc*p*	H	5.18	3.48	3.86	3.63	3.51	3.66	
			C	92.0	71.7	71.3	76.1	74.7	62.5	
	R_β_	→4)-β-D-Glc*p*	H	4.58	3.20	3.69	3.80	3.55	3.66	
			C	95.8	74.1	72.9	76.9	72.9	62.5	
MSP1-2	A	→2)-α-D-Man*p*-(1→	H	5.20	4.06	3.92	3.67	3.93	3.73	3.84
			C	100.9	78.5	70.3	67.2	73.5	60.9	
	B	→6)-α-D-Man*p*-(1→	H	5.02	4.01	3.91	3.68	3.93	3.79	
			C	102.3	70.3	70.6	66.8	73.3	66.6	
	C	→2,6)-α-D-Man*p*-(1→	H	5.08	4.02	3.89	3.69	3.89	3.77	
			C	98.4	78.6	69.0	66.1	74.6	66.5	
	D	α-D-Glc*p*-(1→	H	4.93	3.53	3.67	3.39	3.62	3.73	3.82
			C	98.7	73.3	72.9	69.4	74.2	60.7	
	E	→4)-α-D-Glc*p*-(1→	H	5.36	3.58	3.93	3.61	3.81	3.83	3.71
			C	99.7	71.7	73.4	77.1	71.1	60.6	
	F	→4,6)-α-D-Glc*p*-(1→	H	5.31	3.57	3.92	3.62	3.93	3.66	
			C	99.9	71.8	73.3	77.2	70.3	67.2	
	G	→5)-β-D-Gal*f*-(1→	H	5.34	4.11	4.00	3.95	3.9	3.65	3.80
			C	108.4	81.5	78.7	82.7	78.0	62.7	
	H	β-D-Gal*f*-(1→	H	5.18	4.13	4.00	4.02	3.82	3.66	3.80
			C	109.5	81.6	79.8	82.6	72.4	62.7	

^a^ The overlapping resonances in the table may be interchangeable. a and b represent the chemical shift signals of the two hydrogens at position 6.

**Table 3 foods-11-00982-t003:** Molecular parameters of MSP1-1 and MSP1-2 derived from Berry plots.

Sample	A_2_ (cm^3^mol/g^2^)	Mw (Da)	*R*_g_ (nm)	*R*_h_ (nm)	ρ (*R*_g_/*R*_h_)
MSP1-1	5.4 × 10^−^^3^	1.37 × 10^5^	54.6	28.9	1.89
MSP1-2	−4.5 × 10^−3^	1.65 × 10^4^	51.0	34.9	1.46

## Data Availability

Data is contained within the article.

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
