# Peer review of "Structure Identification of Two Polysaccharides from Morchella sextelata with Antioxidant Activity"

_foods, 2022, doi:10.3390/foods11070982_

Round 1

Reviewer 1 Report

This paper reports on the structure and antioxidant activity of two polysaccharides extracted from Morchella sextelata.

The data on the composition and structure of the polysaccharides appears to be comprehensive and well conducted. The methylation analysis data provided in the supplementary material shows that the derivatives are identified correctly. However, I am concerned about how you have distinguished between the different sugars in the linkage analysis? Depending on the GC column used (which must be provided in the Materials and Methods) several derivatives co-elute; on an HP-5 column, that is commonly used, terminal Man/Glc (in our experience terminal Glc elutes slightly before terminal Man – could the shoulder on peak 1 (FigS3B) be terminal Glc and the main peak terminal Man?) , 2-Man/Glc, 6-Man/Glc, 2,6-Man/Glc either co-elute or elute very close to each other. How have you distinguished these? I would also question the accuracy of the methylation analysis data provided; is it really possible to determine the relative amounts to two decimal places (I suggest that one decimal place is the best you can achieve)?

With regards to the NMR spectroscopy data, I find it hard to believe that it is possible to assign all of the C/H resonances in such complex, overlapped spectra. Assigning the anomeric resonances is reasonable, once the linkages have been identified unequivocally, and 2D spectra may enable some additional resonances to be identified (often tentatively). I am also concerned that you apparently see reducing-end resonances for MSP1-1 when the molecular weight is estimated to be 389 kDa, over 2000 glucose residues. This would suggest a much lower molecular weight polysaccharide, the presence of oligosaccharides, or degradation during analysis.

The proposed structure for polysaccharide MSP1-2 needs to be treated with more circumspect. Whilst HPGFC gives a single peak there still could be more than one polysaccharide type present. The linkage analysis data could be interpreted as being consistent with the presence of glucan, similar to that in MSP1-1, and galactomannan similar to other fungal galactomannans.

Overall, I do not consider that the “detailed structural features” (see Introduction) of the polysaccharides have been identified.

With regards to the antioxidant activity, whilst the polysaccharides do not appear to contain protein or nucleic acids, what other compounds might be present? The monosaccharide data only gives the proportions of the different sugars detected. The sugar data should be provided as mg/g extract, or at least some measure of total carbohydrate content provided.

The carbohydrate nomenclature needs attention. The anomeric and absolute configuration must be adjacent; thus (1→4)-α-D-glucan is correct, with the D/L- configuration given in a smaller font than the main text.

Abstract: I do not understand the end of the first sentence and suggest that you simply say that, “Mushrooms of the Morchella genus exhibit a variety of biological activities.”

Line 15: change to read: …heteropolysaccharide comprised mostly of…

Line 18: change to read: …been observed in a polysaccharide from M. sextelata…

Introduction:

Line 37: change to read: …of polysaccharides is important to understand their activities.

Line 39: this refers to several Morchella species and therefore you need use the plural; i.e. are edible and medicinal mushrooms that are valued for their unique flavor and are categorized as a morphologically recognizable mushrooms.

Line 41: are you referring to one specific species, or the Morchella genus in general?

Line 43: references 4-7 refer to two different species of Morchella; please distinguish which species is being referred to in each case.

Line 46: as above, please refer to the Morchella species being referred to.

Materials and Methods:

Line 62: please clarify how the polysaccharides were fractionated using DEAE-Sepharose. Did you elute only with distilled water and not with increasing salt concentrations? I assume that the column was eluted with more than 6 mL water, but that you collected 6 mL fractions? What was the volume of water used for elution; did you continue to elute with water until no more carbohydrate was detected in the fractions?

Line 67: as above, I assume that you eluted the Sepharose CL-6B column with more than 3 mL, but you collected 3 mL fractions. Please correct this.

Line 83: change to read: Constituent sugar analysis of the polysaccharide fractions….

Line 84: please provide the dimensions of the PA20 column.

Line 90: how were the sugars eluted from the column identified and how were they quantified?

Line 93: change pre-methylated to permethylated.

Line 95: what GC column was used to separate the derivatives and how were they identified and quantified?

Line 98: Please provide more details of what NMR data (check spelling) was collected. Do not repeat sentences, but provide more information on the experimental conditions and data collection.

Line 103: the first sentence does not make sense. What are you measuring using the Congo red test, please specify.

Line 116: state what the experiment is before describing what you did. Thus, “The chain conformation of the polysaccharides was assessed using TEM on a Tecnai G2 Spirit Bio TEM…”

Results and discussion:

Line 134: for the separation of MSP1-1 and 1-2 I suggest that you show on Fig S1B which fractions were pooled.

Line 137: rewrite this to: HPGFC of MSP1-1 and MSP1-2 both gave single peaks indicating that they were homogeneous.

Line 138: rewrite: The molecular weights of MSP1-1 and MSP1-2 were calculated as 389 kDa and 23.4 kDa, respectively, compared with the retention times of dextran standards.

Line 139: rewrite: Constituent sugar analyses showed that MSP1-1 contained only glucose and MSP1-2 contained glucosamine (0.5 mol%), galactose (16.0 mol%), glucose (46.0 mol%), and mannose (37.5 mol%) (Fig. 1B). Also, the legend for Figure 1 please be consistent with your labelling – thus Fucose, Galactosamine, Rhamnose (capital R), Glucosamine (you will not observe the N-acetyl species), Mannuronic acid (not mannose acid).

Author Response

Point 1: The data on the composition and structure of the polysaccharides appears to be comprehensive and well conducted. The methylation analysis data provided in the supplementary material shows that the derivatives are identified correctly.

Response : Thank you for your affirmation of our work.

However, I am concerned about how you have distinguished between the different sugars in the linkage analysis. Depending on the GC column used (which must be provided in the Materials and Methods) several derivatives co-elute; on an HP-5 column, that is commonly used, terminal Man/Glc (in our experience terminal Glc elutes slightly before terminal Man – could the shoulder on peak 1 (FigS3B) be terminal Glc and the main peak terminal Man?) , 2-Man/Glc, 6-Man/Glc, 2,6-Man/Glc either co-elute or elute very close to each other. How have you distinguished these?

Response : Regarding the question of how to distinguish different glycans in methylation analysis, our answer is based mainly on the time of peak emergence. Information on the GC column has been added in Materials and Methods as “BPX70 (25 m × 0.22 mm × 0.25 μm, Trajan) column” (See revised manuscript, main text file, Lines 112–113). Considering the structural similarity of the terminal Glc and the terminal Man and the same MW of the compound after derivatization, the structural similarity leads to a virtually asymptotic peak emergence time (also seen in Figure S3B), which causes a small local overlap and the formation of a shoulder peak. Moreover, the subtle effect of some impurities on the formation of the shoulder peaks cannot be disregarded. In addition, the sequence of peaks is determined under fixed conditions of separation, and we have confirmed through extensive experiments that the terminal Man elutes slightly earlier than the terminal Glc. This finding is consistent with the conclusions obtained by other researchers (see the reference paper "Determining the polysaccharide Composition of plant cell walls" in "Supplementary Table 1")

  • Pettolino, F.A.; Walsh, C.; Fincher, G.B.; Bacic, A. Determining the polysaccharide composition of plant cell walls. Protoc. 2012, 7, 1590-1607.

2-Man/Glc, 6-Man/Glc, and 2,6-Man/Glc were judged mainly according to the order of peak emergence. In addition, the results are checked against the monosaccharide composition results to verify the accuracy and consistency of the data.

I would also question the accuracy of the methylation analysis data provided; is it really possible to determine the relative amounts to two decimal places (I suggest that one decimal place is the best you can achieve)?

 Response : I agree with the reviewer regarding the relative amounts to one decimal place. Hence, I have revised this part of data to retain one decimal place. (See revised manuscript, main text file, Table 1)

Point 2: With regards to the NMR spectroscopy data, I find it hard to believe that it is possible to assign all of the C/H resonances in such complex, overlapped spectra. Assigning the anomeric resonances is reasonable, once the linkages have been identified unequivocally, and 2D spectra may enable some additional resonances to be identified (often tentatively). I am also concerned that you apparently see reducing-end resonances for MSP1-1 when the molecular weight is estimated to be 389 kDa, over 2000 glucose residues. This would suggest a much lower molecular weight polysaccharide, the presence of oligosaccharides, or degradation during analysis.

Response 2: Indeed, it is difficult to assign all C/H resonances using NMR spectra alone, and the overlapping signal peaks are difficult to resolve. We analyzed the NMR signals mainly by comprehensively considering the monosaccharide analysis, methylation results, and the NMR signals already reported in the literature. It is hoped that high magnetic field (800 MHz, 900 MHz) NMR can be used to resolve polysaccharides in the future. The reducing terminal of MSP1-1 was observed in the NMR spectra, and a similar structure was reported for α-glucan from Agaricus blazei Murill and Morchella importuna.

  • Zhang, A.; Deng, J.; Liu, X.; He, P.; He, L.; Zhang, F.; Linhardt, R.J.; Sun, P. Structure and conformation of α-glucan extracted from Agaricus blazei Murill by high-speed shearing homogenization. J. Biol. Macromol. 2018, 113, 558-564.
  • Wen, Y.; Bi, S.; Hu, X.; Yang, J.; Li, C.; Li, H.; Yu, D.B.; Zhu, J.; Song, L.; Yu, R. Structural characterization and immunomodulatory mechanisms of two novel glucans from Morchella importuna fruiting bodies. J. Biol. Macromol. 2021, 183, 145-157.

Based on the reviewer's suggestions, Figure 3G was revised.

Point 3: The proposed structure for polysaccharide MSP1-2 needs to be treated with more circumspect. Whilst HPGFC gives a single peak there still could be more than one polysaccharide type present. The linkage analysis data could be interpreted as being consistent with the presence of glucan, similar to that in MSP1-1, and galactomannan similar to other fungal galactomannans.

Overall, I do not consider that the “detailed structural features” (see Introduction) of the polysaccharides have been identified.

 Response 3: Thanks to the reviewer's suggestion, we have re-analyzed the NMR spectra of MSP1-2 and referred to the literature for the NMR analysis of polysaccharides structures in similar situations.

  • Choma, A.; Nowak, K.; Komaniecka, I.; WaÅ›ko, A.; PleszczyÅ„ska, M.; Siwulski, M.; Wiater, A. Chemical characterization of alkali-soluble polysaccharides isolated from a Boletus edulis (bull.) fruiting body and their potential for heavy metal bio-sorption. Food Chem. 2018, 266, 329-334.
  • Dimopoulou, M.; Alba, K.; Sims, I.M.; Kontogiorgos, V. Structure and rheology of pectic polysaccharides from baobab fruit and leaves. Carbohyd. Polym. 2021, 273, 118540.

I agree with the reviewer's suggestion that MSP1-2 was galactomannan with co-extracted α-glucan. This part has been revised as follows:

The HMBC (Figure 4E) and NOESY spectra (Figure 4F) identify sequences between sugar residues in MSP1-2. A cross-peak between D H-1 and F H-6 indicates that sugar residue D is connected to sugar residue F by an α-1,6 glycosidic bond. Observed cross-peak of G H-1/C H-6 implies the existence of G→C linkage. In addition, the interaction between B H-1/C H-6 confirms the B→C linkage. Similarly, the observed cross-peaks of A H-1/B C-6, F H-1/E C-4, and H H-1/G C-5 confirm the A→B, F→E, H→G linkage. Overall, we infer that MSP1-2 is a polymer with a skeleton consisting of 1,2-, 1,6-, 1,2,6-linked α-D-Manp residues and (1→4)-α-D-glucose. The possible sequence fragments present in MSP1-2 are shown in Figure 4G. (See revised manuscript, main text file, Lines 228–236).

The “detailed structural features” has been revisied to “structural features”. (See revised manuscript, main text file, Line 60).

Point 4: With regards to the antioxidant activity, whilst the polysaccharides do not appear to contain protein or nucleic acids, what other compounds might be present? The monosaccharide data only gives the proportions of the different sugars detected. The sugar data should be provided as mg/g extract, or at least some measure of total carbohydrate content provided.

Response 4: We apologize for the confusing expression. Other compounds may be present in MSP1-1 or MSP1-2, but carbohydrate is the predominant compound. The total carbohydrate contents of MSP1-1 and MSP1-2 were 98.02% ± 0.80% and 98.68% ± 0.69%, respectively. This part of the data has been added to the article. (See revised manuscript, main text file, Lines 162–163).

Point 5: The carbohydrate nomenclature needs attention. The anomeric and absolute configuration must be adjacent; thus (1→4)-α-D-glucan is correct, with the D/L- configuration given in a smaller font than the main text.

Response 5: All glycosidic bond formats have been checked, and the font for the D/L-configuration has been chosen to be smaller than the main text at 8 point.

Point 6: Abstract: I do not understand the end of the first sentence and suggest that you simply say that, “Mushrooms of the Morchella genus exhibit a variety of biological activities.”

Response : This was corrected. (See revised manuscript, main text file, Line 10).

Line 15: change to read: …heteropolysaccharide comprised mostly of…

Response : This was corrected. (See revised manuscript, main text file, Line 14).

Line 18: change to read: …been observed in a polysaccharide from M. sextelata…

Response : This was corrected. (See revised manuscript, main text file, Lines 16–17).

Point 7: Introduction: Line 37: change to read: …of polysaccharides is important to understand their activities.

Response : This was corrected. (See revised manuscript, main text file, Lines 45–46).

Line 39: this refers to several Morchella species and therefore you need use the plural; i.e. are edible and medicinal mushrooms that are valued for their unique flavor and are categorized as a morphologically recognizable mushrooms.

Response : This was corrected. (See revised manuscript, main text file, Lines 48–49).

Line 41: are you referring to one specific species, or the Morchella genus in general?

Response :Morchella” was expressed as the Morchella genus in general. We apologize for the confusion; this has been replaced with “morel”. (See revised manuscript, main text file, Line 51).

Line 43: references 4-7 refer to two different species of Morchella; please distinguish which species is being referred to in each case.

Response : Examples of references 4-7 are intended to express the biological activity possessed by this population of morel mushrooms. “Morchella” has been corrected to “morel”. (See revised manuscript, main text file, Line 53).

Line 46: as above, please refer to the Morchella species being referred to.

Response : Examples of references 5,8-10 are intended to express the structures possessed by this population of morel mushrooms. “Morchella” has been corrected to “morel”. (See revised manuscript, main text file, Line 55).

Point 8: Materials and Methods:

Line 62: please clarify how the polysaccharides were fractionated using DEAE-Sepharose. Did you elute only with distilled water and not with increasing salt concentrations? I assume that the column was eluted with more than 6 mL water, but that you collected 6 mL fractions? What was the volume of water used for elution; did you continue to elute with water until no more carbohydrate was detected in the fractions?

Response : We apologize for the confusion. Yes, the polysaccharide fraction was eluted using distilled water and not increasing salt concentrations. The 6 mL refers to the volume of eluate per tube, and a total of 600 mL of distilled water was used to elute until no carbohydrates were detected. This part has been rewritten as follows:

The MSPSWE (20 mg/mL) was introduced into a DEAE-Sepharose FF (2.6 × 60 cm) column and subsequently washed with running distilled water (flow rate: 1.2 mL/min; liquid volume: 6 mL/tube). The carbohydrate content in the tubes was monitored. The main frac-tion (MSP1) was combined and freeze-dried. Then, MSP1 (6 mg/mL) was further eluted with distilled water (flow rate: 0.3 mL/min; liquid volume: 3 mL/tube) on a Sepharose CL-6B (1.6 × 90 cm) column. The two fractions derived from MSP1 were labeled MSP1-1 and MSP1-2. (See revised manuscript, main text file, Lines 71–77).

Line 67: as above, I assume that you eluted the Sepharose CL-6B column with more than 3 mL, but you collected 3 mL fractions. Please correct this.

Response : The 3 mL refers to the volume of eluate per tube. This part has been rewritten. (See revised manuscript, main text file, Lines 71–77).

Line 83: change to read: Constituent sugar analysis of the polysaccharide fractions….

Response : This was corrected. (See revised manuscript, main text file, Line 91).

Line 84: please provide the dimensions of the PA20 column.

Response : This was corrected. “PA20 (150 mm × 3.0 mm id) column”. (See revised manuscript, main text file, Line 93)

Line 90: how were the sugars eluted from the column identified and how were they quantified?

Response : High-performance anion exchange chromatography coupled with pulsed amperometric detection was used to determine the monosaccharide composition. The monosaccharide species were characterized according to retention time against the sugar standard curve. Through precise configuration of each monosaccharide standard solution (5 mg/L) according to the single standard method, the concentration of different monosaccharides was determined, and the molar ratio was calculated according to the molar mass of monosaccharides.

Line 93: change pre-methylated to permethylated.

Response : This was corrected. (See revised manuscript, main text file, Line 101).

Line 95: what GC column was used to separate the derivatives and how were they identified and quantified?

Response : Information about the GC column has been added in Materials and Methods as BPX70(25 m × 0.22 mm × 0.25 μm, Trajan) column. The characterization is mainly based on the retention time of the target compound on the analytical column. With the elution conditions determined, different compounds exhibit different retention capacity on the analytical column, and the elution time consists of many differences. By identifying the secondary mass spectrometry fragments and comparing them with the existing database, the bonding mode is thus confirmed. Quantification is mainly based on the peak area for calculating the relative molar ratio of each residue.

Line 98: Please provide more details of what NMR data (check spelling) was collected. Do not repeat sentences, but provide more information on the experimental conditions and data collection.

Response : Thanks to the reviewer's suggestion, the section on NMR in the methods has been revised as follows: The NMR data were obtained using a 500 MHz Bruker NMR spectrometer (Zurich, Switzerland) at 25 °C. MSP1-1 and MSP1-2 (20 mg each) were dissolved in D2O solvent and lyophilized. The dried polysaccharides (20 mg) were transferred to a standard 5 mm probe after dissolution in 500 μL D2O solvent. The operating frequen-cies of 1H NMR and 13C NMR were 500 MHz and 125 MHz, respectively. The acquisi-tion times were set to 16 times for 1H NMR spectra, 1024 times for 13C NMR spectra, 12 times for COSY spectra, 12 times for HSQC spectra, 32 times for NOESY spectra, and 64 times for HMBC spectra. MestReNova software (Version: 9.1.0-14011) was used to process the NMR data of the polysaccharide samples. (See revised manuscript, main text file, Lines 117–125).

Line 103: the first sentence does not make sense. What are you measuring using the Congo red test, please specify.

Response : The first sentence has been rewritten. The test confirming whether the polysaccharide samples had a helical structure was conducted using a method described by Jia et al. (See revised manuscript, main text file, Lines 127–128).

Line 116: state what the experiment is before describing what you did. Thus, “The chain conformation of the polysaccharides was assessed using TEM on a Tecnai G2 Spirit Bio TEM…”

Response: This was corrected.

The chain conformation of the polysaccharides was assessed using a Tecnai G2 Spirit Bio TEM (Thermo Fisher Scientific Inc., USA) at 80 kV. MSP1-1 or MSP1-2 was dissolved in 0.9% NaCl solution to achieve a final concentration of 5 μg/mL. A drop of the polysaccharide solution was deposited on the support carbon film (200 mesh), dried in the air, and stained with 0.2% phosphotungstic acid. (See revised manuscript, main text file, Lines 141–145).

Point 9: Results and discussion:

Line 134: for the separation of MSP1-1 and 1-2 I suggest that you show on Fig S1B which fractions were pooled.

Response: Thanks to the reviewer for the suggestion. This has been marked in the Fig S1B.

Line 137: rewrite this to: HPGFC of MSP1-1 and MSP1-2 both gave single peaks indicating that they were homogeneous.

Response: This was corrected. (See revised manuscript, main text file, Lines 166–167).

Line 138: rewrite: The molecular weights of MSP1-1 and MSP1-2 were calculated as 389 kDa and 23.4 kDa, respectively, compared with the retention times of dextran standards.

Response: This was corrected. (See revised manuscript, main text file, Lines 167–168).

Line 139: rewrite: Constituent sugar analyses showed that MSP1-1 contained only glucose and MSP1-2 contained glucosamine (0.5 mol%), galactose (16.0 mol%), glucose (46.0 mol%), and mannose (37.5 mol%) (Fig. 1B). Also, the legend for Figure 1 please be consistent with your labelling – thus Fucose, Galactosamine, Rhamnose (capital R), Glucosamine (you will not observe the N-acetyl species), Mannuronic acid (not mannose acid).

Response: These were corrected. (See revised manuscript, main text file, Lines 25–27; 168–171).

Reviewer 2 Report

Please reduce the repetitive rate

Author Should add rationale in the end of introduction.

Methods should be more explanatory and author could add a list of abbreviations.

Author should clearly mention the design of statistical analysis.

Overall article is good and could be published as it will increase the scientific information of the reader.

Author Response

Point 1: Please reduce the repetitive rate.

Response 1: Thank you for your kind suggestion. We have revised the sentences with high repetition rate in the article.

Point 2: Author Should add rationale in the end of introduction.

Response 2: Thank you for your suggestion. The rationale has been added in the end of the introduction.

This study aimed to obtain the detailed structural features of purified MSPSWE. Moreover, the in vitro antioxidant activities of polysaccharides were evaluated. These results will be helpful for elucidating the structure–activity relationships of M. sextelata polysaccharides and promote their application in functional foods. (See revised manuscript, main text file, Lines 60–63).

Point 3: Methods should be more explanatory and author could add a list of abbreviations.

Response 3: Methods have been modified, and the specific revisions are as follows:

The MSPSWE (20 mg/mL) was introduced into a DEAE-Sepharose FF (2.6 × 60 cm) column and subsequently washed with running distilled water (flow rate: 1.2 mL/min; liquid volume: 6 mL/tube). The carbohydrate content in the tubes was monitored. Then, the main fraction (MSP1) was combined and freeze-dried. MSP1 (6 mg/mL) was further eluted with distilled water (flow rate: 0.3 mL/min; liquid volume: 3 mL/tube) on a Sepharose CL-6B (1.6 × 90 cm) column. The two fractions de-rived from MSP1 were labeled MSP1-1 and MSP1-2. (See revised manuscript, main text file, Lines 71–77).                                                                                       

UV–vis spectroscopy was performed on the purified polysaccharides (1.0 mg/mL each) using a Shimadzu UV-2550 spectrophotometer (Shimadzu, Japan). The scan-related parameters were set as scan interval: 0.5 nm and scan range: 200–400 nm. (See revised manuscript, main text file, Lines 85–87).

MSP1-1 and MSP1-2 were permethylated according to the following steps: firstly, 10 mg of MSP1-1 or MSP1-2 was weighed and dissolved in 1 mL of ultrapure water, and equal volumes of carbodiimide (100 mg/mL), imidazole (2 M), and NaBD4 (30 mg/mL) were drawn to react with the sample solution. After 3 h, the reaction was subsequently terminated by the addition of glacial acetic acid. The polysaccharide sample was then dialyzed, lyophilized and dissolved in dimethyl sulfoxide, and the solution was added to a NaOH solution (50 μL) and reacted with methyl iodide for 1 h of methylation. The product was subsequently extracted using dichloromethane and then dried with nitrogen. Trifluoroacetic acid (2 M) was added to the product after methylation was completed, and a hydrolysis procedure was performed (reaction at 121 °C for 90 min). Finally, the obtained product was reduced with 2 M ammonia and 1 M NaBD4 and acetylated with acetic anhydride. To detect the ultimate derivatives, an Agilent 7890A-5977B GC-MS instrument (CA, USA) with a BPX70(25 m × 0.22 mm × 0.25 μm, Trajan)column was employed with high-purity helium as the carrier gas at an injection volume of 1 μL. The temperature of the BPX70 column was initially 140 °C (held for 2.0 min) and then ramped up to 230 °C (held for 3 min) using a 3 °C /min program. (See revised manuscript, main text file, Lines 100–115).

A list of abbreviations has been added. (See revised manuscript, main text file, Lines 22–32).

Point 4: Author should clearly mention the design of statistical analysis.

Response 4: We’re sorry for neglecting to express the design of the specific statistical analysis. The section has been entirely rewritten as follows:

All data were obtained using three independent trials, and results were expressed as mean ± standard deviation. One-way analysis of variance (ANOVA) was performed using SPSS 18.0 software. Tukey test results were significantly different in cases where the p-values were less than 0.05. (See revised manuscript, main text file, Lines 155–158).

Point 5: Overall article is good and could be published as it will increase the scientific information of the reader.

Response 5: Thank you for your recognition of our work. All your suggestions are very important and will guide us in my future research work. Thank you again for your advice!

Round 2

Reviewer 1 Report

This revised paper has clarified many questions, although further changes are required. I suggest that the manuscript is edited by an English language expert to improve the readability of the paper; some comments are given below.

Line 11: change derived to isolated.

Line 18: I question whether the polysaccharide have excellent antioxidant activity as implied in this sentence; please rewrite this.

Line 19: the last sentence says nothing; it provided evidence for the development M. sextelata polysaccharides for what?

Line 51: please check the punctuation in this sentence.

Line 69: ensure that the species name starts with a small letter, not a capital – M. sextelata.

Line 75: repetition of 6 mg/mL; please delete the second one. How much material (total weight) of sample was applied to the column? Also, I suggest you change to read – “MSP1 (X mg) was fractionated further by gel filtration chromatography on a Sepharose CL-6B (1.6 × 90 cm) column. The sample (6 mg/mL) was applied to the column, eluted with distilled water (flow rate: 0.3 mL/min; 3 mL/fraction) and the two fractions obtained were labelled as MSP1-1 and MSP1-2.

Line 103: what does this mean – “equal volumes of …. were drawn to react”? Do you mean that you added equal volumes of these reagents to activate and reduce uronic acids? Why would you do this when you have specifically isolated only neutral polysaccharides by ion-exchange chromatography, and constituent sugar analysis shows that there are no uronic acids in either of the polysaccharide fractions?

Line 115: how did you identify the derivatives in your methylation analysis?

Line 161: the yields and carbohydrate contents of the fractions is given to high accuracy; as for the methylation data in the original manuscript I suggest that this changed to one decimal place.

Line 169: change this to read: “…MSP1-2 contained mostly glucose (46.0 mol%), mannose (37.5 mol%) and galactose (16.0 mol%), with a small amount of glucosamine (0.5 mol%).”

Line 189: reduce to one decimal place.

Table 1: T-Gal(f) should be 1,4-di-O-acetyl-2,3,5,6-tetra-O-methyl galactitol.

Lines 205 and 223: the results are not summarised and so I suggest you say, “..and given in Table 2.”

Table 2: please show carbon resonances to one decimal place only (and in the text). I also think that the table should have a footnote stating that overlapping resonances may be interchangeable as I think that many of resonances cannot be assigned confidently.

Line 220: what is the specific evidence that the Gal(f) resonances (G and H) are in the β-configuration? See https://doi.org/10.1016/S0065-2318(08)60055-4 for details.

Author Response

Point 1: This revised paper has clarified many questions, although further changes are required. I suggest that the manuscript is edited by an English language expert to improve the readability of the paper; some comments are given below.

Response: Thanks to the reviewer’s suggestions. Accordingly, the manuscript has been edited by an English language expert.

Point 2: Line 11: change derived to isolated.

Response: The word choice has been rectified. (See revised manuscript, main text file, Line 11)

Point 3: Line 18: I question whether the polysaccharide have excellent antioxidant activity as implied in this sentence; please rewrite this.

Response: We apologize for the confusing expression. The sentence has been revised—“both MSP1-1 and MSP1-2 possessed good thermal stability and showed antioxidant activity.” (See revised manuscript, main text file, Line 18)

Point 4: Line 19: the last sentence says nothing; it provided evidence for the development M. sextelata polysaccharides for what?

Response: We apologize for the confusing expression. The sentence has been revised as “This study provides fundamental data on the structure–activity relationships of M. sextelata polysaccharides. ” (See revised manuscript, main text file, Line 18–19)

Point 5: Line 51: please check the punctuation in this sentence.

Response: The punctuation error has been corrected. (See revised manuscript, main text file, Line 52)

Point 6: Line 69: ensure that the species name starts with a small letter, not a capital – M. sextelata.

Response: All species names in the manuscript have been set to lowercase, as suggested by the reviewer.

Point 7: Line 75: repetition of 6 mg/mL; please delete the second one. How much material (total weight) of sample was applied to the column? Also, I suggest you change to read – “MSP1 (X mg) was fractionated further by gel filtration chromatography on a Sepharose CL-6B (1.6 × 90 cm) column. The sample (6 mg/mL) was applied to the column, eluted with distilled water (flow rate: 0.3 mL/min; 3 mL/fraction) and the two fractions obtained were labelled as MSP1-1 and MSP1-2.

Response : We apologize for the confusing expression. The repetition has been rectified. (See revised manuscript, main text file, Lines 75–79)

Point 8: Line 103: what does this mean – “equal volumes of …. were drawn to react”? Do you mean that you added equal volumes of these reagents to activate and reduce uronic acids? Why would you do this when you have specifically isolated only neutral polysaccharides by ion-exchange chromatography, and constituent sugar analysis shows that there are no uronic acids in either of the polysaccharide fractions?

Response: Thanks to the reviewer's suggestion. The nonclarity was due to our writing; hence, this section has been revised for clarity.

“MSP1-1 and MSP1-2 were permethylated according to the following steps: first, 10 mg of MSP1-1 or MSP1-2 was weighed and dissolved in 500 μL of DMSO, and 50 μL of DMSO/NaOH solution (120 mg/mL) was added to the sample solution to initiate a reaction. After 30 min, iodomethane (50 μL) was added and reacted for 1 h. The reaction was terminated by the addition of deionized water (1 mL). The product was subsequently extracted using dichloromethane (500 μL) and dried with nitrogen. Trifluoroacetic acid (2 M) was added to the product after methylation was completed, and a hydrolysis procedure was performed (reaction at 121 °C for 90 min). Finally, the obtained product was reduced with 2 M ammonia and 1 M NaBD4 and acetylated with acetic anhydride. To detect the ultimate derivatives, an Agilent 7890A-5977B GC-MS instrument (CA, USA) with a BPX70 (25 m × 0.22 mm × 0.25 μm, Trajan) was employed with high-purity heli-um as carrier gas at an injection volume of 1 μL. The temperature of the BPX70 col-umn was initially 140 °C (held for 2.0 min) and then ramped up to 230 °C (held for 3 min) using a 3 °C /min program. The peaks of the derivatives were identified from the relative retention times and mass spectral data.” (See revised manuscript, main text file, Lines 102–117)

Point 9: Line 115: how did you identify the derivatives in your methylation analysis?

Response: The identification of the derivatives has been described—“The peaks of the derivatives were identified by the relative retention times and mass spectral data.” (See revised manuscript, main text file, Line 14).

Point 10: Line 161: the yields and carbohydrate contents of the fractions is given to high accuracy; as for the methylation data in the original manuscript I suggest that this changed to one decimal place.

Response: The number of significant figures in the decimals has been corrected. (See revised manuscript, main text file, Line 163, Table 1).

Point 11: Line 169: change this to read: “…MSP1-2 contained mostly glucose (46.0 mol%), mannose (37.5 mol%) and galactose (16.0 mol%), with a small amount of glucosamine (0.5 mol%).”

Response: The statement has been rephrased following your suggestion. (See revised manuscript, main text file, Lines 170–172).

Point 12: Line 189: reduce to one decimal place.

Response: Each percentage has been reduced to one decimal place. (See revised manuscript, main text file, Line 190).

Point 13: Table 1: T-Gal(f) should be 1,4-di-O-acetyl-2,3,5,6-tetra-O-methyl galactitol.

Response : We apologize for the confusion. The name of the molecule has been corrected. (See revised manuscript, main text file, Table 1).

Point 14: Lines 205 and 223: the results are not summarised and so I suggest you say, “..and given in Table 2.”

Response: The suggested phrasing has been inserted at the relevant instances. (See revised manuscript, main text file, Line 206 and 224).

Point 15: Table 2: please show carbon resonances to one decimal place only (and in the text). I also think that the table should have a footnote stating that overlapping resonances may be interchangeable as I think that many of resonances cannot be assigned confidently.

Response: The decimal places have been reduced, and the suggested statement has been added in the footnote of the table. (See revised manuscript, main text file, Table 2).

Point 16: Line 220: what is the specific evidence that the Gal(f) resonances (G and H) are in the β-configuration? See https://doi.org/10.1016/S0065-2318(08)60055-4 for details.

Response: The inference that the Gal(f) resonance (G and H) are in the β-configuration has been drawn on the basis of literature. The pertinent papers have been cited. (See revised manuscript, main text file, Lines 221 and 384).
